# Botulinum Toxin Complex Serotype B-Okra Exerts Systemic Toxicity via the Oral Route by Disrupting the Intestinal Epithelial Barrier

**DOI:** 10.3390/toxins17090443

**Published:** 2025-09-04

**Authors:** Chiyono Morimoto, Sho Amatsu, Takuhiro Matsumura, Masahiko Zuka, Yukako Fujinaga

**Affiliations:** 1Department of Bacteriology, Graduate School of Medical Sciences, Kanazawa University, Kanazawa 920-8640, Japant-matsu@med.kanazawa-u.ac.jp (T.M.); 2Department of Forensic Medicine and Pathology, Graduate School of Medical Sciences, Kanazawa University, Kanazawa 920-8640, Japan

**Keywords:** *Clostridium botulinum*, botulinum toxin, botulinum toxin complex, in vitro reconstitution, hemagglutinin, E-cadherin, epithelial barrier

## Abstract

Botulinum toxin (BoNT) causes flaccid paralysis by blocking the release of neurotransmitters. BoNTs associate with neurotoxin-associated proteins to form medium and large progenitor toxin complexes. The large progenitor toxin complex serotype A-62A (L-PTC/A-62A) specifically targets intestinal M cells for invasion, whereas large progenitor toxin complex serotype B-Okra (L-PTC/B-Okra) is mainly taken up by enterocytes and exhibits higher toxicity via the oral route. Hemagglutinin (HA) is a neurotoxin-associated protein that promotes BoNT absorption from the intestine and has carbohydrate-binding and barrier-disrupting activities. In this study, we established an in vitro reconstitution and purification system for recombinant L-PTC/B-Okra and created a recombinant L-PTC/B-Okra mutant rL-PTC/B-KA with carbohydrate-binding activity but not barrier-disrupting activity. rL-PTC/B-KA showed significantly reduced oral toxicity. Our results demonstrate that the B-Okra toxin disrupts the epithelial barrier of enterocytes and exerts oral toxicity.

## 1. Introduction

Botulinum toxin (BoNT) is the most potent naturally occurring toxin and is responsible for causing botulism, a life-threatening condition characterized by flaccid paralysis. BoNT blocks neurotransmitter release by cleaving soluble *N*-ethylmaleimide-sensitive-factor attachment protein receptor (SNARE) proteins at the presynaptic terminals of neurons [1]. BoNTs are produced by *Clostridium botulinum* and related species as progenitor toxin complexes (PTCs) with neurotoxin-associated proteins (NAPs) [2]. Neurotoxin-associated proteins comprise a non-toxic non-hemagglutinin (NTNHA) and three hemagglutinins (HAs): HA1, HA2, and HA3 (also known as HA33, HA17, and HA70, respectively). NTNHA associates with BoNT to form a medium progenitor toxin complex (M-PTC) and protects the toxin from digestion and destabilization in gastrointestinal juice [2,3]. The HA complex is assembled from six HA1, three HA2, and three HA3 [4,5,6,7], and associates with M-PTC to form a large progenitor toxin complex (L-PTC), which exhibits approximately 700-fold higher oral toxicity than M-PTC [2,8]. HA facilitates the intestinal absorption of BoNTs through at least two activities: carbohydrate-binding [5,9,10,11,12,13,14] and barrier-disruption [13,14,15,16,17,18,19]. L-PTC binds to the luminal surface of the intestinal epithelial cells via HA’s carbohydrate-binding activity [9,20,21,22,23,24,25]. After transcytosis from the apical to the basolateral surface, HA binds to E-cadherin and inhibits cell–cell adhesion; HA disrupts the epithelial barrier, resulting in the paracellular transport of BoNT across the intestine [15,16,18]. In HA-negative strains, OrfX genes are associated with the BoNT gene instead of HA genes [26]. It has recently been reported that OrfX2 can assemble with M-PTC and boost the oral toxicity [27], although the mechanism remains largely unknown.

BoNTs have been classified into seven serotypes (A–G), of which serotypes A, B, E, and F cause human botulism [28]. L-PTCs are classified as “hyper-oral-toxic (HOT)” or “non-HOT” based on their relative oral toxicity levels [2,8,25]. These classes can be categorized based on the carbohydrate-binding activity of HA [25]. L-PTC serotype A-62A (L-PTC/A-62A), a non-hyper-oral-toxic toxin, targets intestinal microfold (M) cells for entry [23]. Intestinal M cells are specialized epithelial cells with the ability to transport macromolecules and microorganisms from the gut lumen to the underlying lymphoid tissue to induce antigen-specific immune response and are located in the follicle-associated epithelium (FAE) of Peyer’s patches [29]. L-PTC/A-62A exerts oral toxicity by disrupting the epithelial barrier around M cells [18,23]. In contrast, L-PTC serotype B-Okra (L-PTC/B-Okra), which exhibits 20–80-fold greater oral toxicity and is classified as HOT-type, is taken up by enterocytes (also known as intestinal absorptive cells) [25].

Enterocytes are the most abundant epithelial cell lineage in the small and large intestine, and especially constitute more than 80% of all epithelial cells in the small intestine [30]. A single layer of enterocytes lining the intestinal epithelium serves as the physical and functional barrier between the luminal contents of the gut and the host [31]. The HA of L-PTC/B-Okra increases the intestinal permeability of fluorescein isothiocyanate (FITC)-dextran and PTCs [15], although the effect of the barrier-disrupting activity of L-PTC/B-Okra on oral toxicity remains unclear. In this study, we established an in vitro reconstitution and purification system for recombinant L-PTC/B-Okra and created a recombinant L-PTC/B-Okra mutant with carbohydrate-binding activity, but not barrier-disrupting activity (rL-PTC/B-KA). Our results demonstrate that the barrier-disrupting activity of the HOT-type toxin L-PTC/B-Okra is critically involved in its toxicity via the oral route.

## 2. Results

### 2.1. In Vitro Reconstitution and Purification of Recombinant L-PTC (rL-PTC)

To elucidate the impact of HA differences between L-PTC/B and L-PTC/A on systemic toxicity via the oral route, we created wild-type rL-PTC/B (rL-PTC/B-WT) and a chimeric L-PTC/B comprising HA/A instead of HA/B (rL-PTC/BA-WT) (Figure 1 and Figure 2A–C) [25]. HA3 Lys607 is crucial for the binding of HA to E-cadherin, and its alanine substitution (K607A, KA) impairs E-cadherin binding and barrier-disrupting activities [17]. We also generated rL-PTCs containing HA3-K607A (rL-PTC/B-KA and rL-PTC/BA-KA) (Figure 2B,C). These rL-PTCs contained similar amounts of BoNT/B, NTNHA/B, and HAs (Figure 2B,C).

### 2.2. Carbohydrate-Binding Activity

L-PTC binds to the intestinal epithelium through the carbohydrate-binding activity of HA [9,20,21,22,23,24,25]. We assessed the carbohydrate-binding activity of rL-PTCs by ELISA using porcine gastric mucin (PGM) and bovine submaxillary mucin (BSM), which mediate galactose- and sialic acid-dependent binding, respectively [17]. rL-PTC/B-WT and rL-PTC/B-KA bound to PGM and BSM in a similar manner of nL-PTC/B (Figure 3). HA/A has more extended carbohydrate-binding pockets than HA/B, and both native L-PTC/A and rHA/A bind mucins with higher affinity than native L-PTC/B and rHA/B [25]. Consistent with this, the chimera toxins rL-PTC/BA-WT and rL-PTC/BA-KA, in which HA/B is replaced with HA/A, bind mucins in a similar manner of native L-PTC/A and rHA/A (Figure 3). These data show that swapping HA in the rL-PTCs alters the carbohydrate-binding spectrum, and that the HA3 K607A mutation does not impair the carbohydrate-binding activities.

### 2.3. Epithelial Barrier-Disrupting Activity

HA/A and HA/B inhibit cell–cell adhesion in epithelial cells by binding to E-cadherin, resulting in epithelial barrier disruption [13,14,15,16,17,18,19]. HA2 and HA3^mini^ (aa 380-626) are responsible for the binding of HA to E-cadherin. Notably, the recombinant HA/B harboring the HA3-K607A mutant (rHA/B-KA) is deficient in barrier-disrupting activity [17,18,19]. Consistent with this, rL-PTC/B-WT and rL-PTC/BA-WT disrupted the epithelial barrier of Caco-2 cell monolayers and exhibited barrier-disrupting activity comparable to that of native L-PTC/B and rHA/A, respectively (Figure 4). These data demonstrate that the wild-type recombinant L-PTCs are fully functional in disrupting the epithelial barrier. In contrast, rL-PTC/B-KA and rL-PTC/BA-KA had no effect on the epithelial barrier function (Figure 4), indicating that the HA3 K607A mutation impairs the barrier-disrupting activities of L-PTCs.

### 2.4. Toxicities of rL-PTC

The intraperitoneal toxicity of rL-PTCs was assessed by intraperitoneal (i.p.) administration of 200 pg rL-PTCs to mice. L-PTCs absorbed from the intestines or injected into the bloodstream dissociate into BoNT and neurotoxin-associated proteins in the circulation [2,3]. In other words, intraperitoneal injection resulted in the toxicity of BoNT alone in the L-PTC complex. rL-PTC/B-WT, rL-PTC/B-KA, rL-PTC/BA-WT, and rL-PTC/BA-KA all harbored comparable amounts of native BoNT/B (Figure 2) and exhibited similar BoNT toxicity (Figure 5A), indicating that the oral toxicity depends on the intestinal absorption rather than intrinsic BoNT activity.

The oral toxicity of rL-PTCs was assessed by intragastrically (i.g.) administering 200 ng of rL-PTCs/B or 2 μg rL-PTCs/BA to mice. rL-PTC/B-WT exhibited at least a 10-fold higher oral toxicity than rL-PTC/BA-WT, indicating that the difference in HA in L-PTC affected oral toxicity (Figure 5B,C) [25]. rL-PTC/B-KA, which lacks barrier-disrupting activity (Figure 4A), exhibited dramatically reduced toxicity (Figure 5B), similar to that of rL-PTC/BA-KA (Figure 5C). These data demonstrate that barrier-disrupting activity is crucial for the oral toxicity of both serotypes, B-Okra and A-62A.

## 3. Discussion

The large progenitor toxin complex (L-PTC) comprises 14 protein subunits, including BoNT, NTNHA, HA3, HA2, and HA1, in a 1:1:3:3:6 stoichiometry [4,5,6,7]. Consistent with a previous study on recombinant L-PTC/A-62A [18], we established an in vitro reconstitution and purification system for recombinant L-PTCs/B-Okra (rL-PTCs/B) using native BoNT/B-Okra and its recombinant components (Figure 1 and Appendix A). We found that the HA3 K607A mutation significantly reduced the oral toxicity of recombinant L-PTC/B-Okra without affecting BoNT activity and carbohydrate-binding. These findings highlight that barrier-disrupting activity is crucial for oral toxicity.

L-PTC/A-62A enters the host through intestinal microfold (M) cells in Peyer’s patches [23]. Once in the intestine, the neurotoxin-associated proteins of L-PTC/A-62A disrupts the epithelial barrier around M cells [23]. Additionally, Lee et al. [18] reported that the absence of this barrier-disrupting activity significantly impaired the oral toxicity of recombinant L-PTC/A-62A. The HA of L-PTC/A-62A augmented oral toxicity by compromising the barrier integrity of intestinal M cells. In contrast, hyper-oral-toxic (HOT)-type toxin of serotype B-Okra, unlike non-HOT-type toxin of serotype A-62A, enters the host not only through intestinal M cells but also through the absorptive epithelial cells (enterocytes) of the small intestine [25]. We found that the HA of L-PTC/B disrupts the epithelial barrier of enterocytes in the small intestine [15], and the absence of this activity reduces the oral toxicity of recombinant L-PTC/B-Okra. Together, these findings indicate that HA-mediated barrier disruption facilitates paracellular BoNT uptake through M cells and enterocytes, and that this mechanism contributes to efficient intestinal absorption in HOT- and non-HOT-type toxins.

The intestinal epithelial barrier serves as a boundary between the mucosal immune system and the external environment of the gut lumen [31], yet it also presents a unique opportunity for developing oral delivery systems. For example, M cell-mediated uptake facilitates the entry of antigens [32,33,34,35,36], and transferrin conjugation triggers receptor-mediated transcytosis of cargoes across the intestinal epithelium [37]. The C-terminal fragment of *Clostridium perfringens* enterotoxin (C-CPE) binds to the tight junction (TJ) protein claudin [38,39] and modulates the TJ seal, enabling the paracellular absorption of small molecules [40]. Moreover, HA of botulinum toxin complexes from A-62A and B-Okra can disrupt the E-cadherin–mediated cell–cell adhesion, enabling the paracellular absorption of macromolecules [15,16,23]. These findings suggest that by exploiting multiple intestinal uptake routes—such as transcytosis through M cells and enterocytes in the presence or absence of paracellular transport—engineered toxin-based cargo delivery systems could facilitate efficient mucosal vaccines or therapeutic agents.

## 4. Materials and Methods

### 4.1. Plasmid Construction

Genomic DNA was extracted and purified from *C. botulinum* serotype B strain Okra and serotype A strain 62A. NTNHA (aa 1-1197) derived from the serotype B BoNT complex (NTNHA/B)-encoding gene, was cloned into the *Nhe*I-*Sal*I site of the pET28b(+) vector (Merck, Darmstad, Germany) (pET28b-His-NTNHA/B). HA3 (aa 19-626) encoding genes derived from serotype B and A BoNT complexes (HA3/B and HA3/A, respectively) were cloned into the *Kpn*I-*Sal*I site of the pET52b(+) vector (Merck) (pET52b-strep-HA3/B or/A). DNA fragments encoding His-NTNHA/B and strep-HA3s were amplified by PCR and cloned into the *Xba*I-*Sac*I and *Nde*I-*Xho*I sites of pETDuet(+) (Merck) using a GeneArt^TM^ Seamless Cloning and Assembly kit (Thermo Fisher Scientific, Waltham, MA, USA). Site-directed mutagenesis was performed using the PrimeSTAR Max Polymerase (TaKaRa Bio, Shiga, Japan). The inserted regions of these plasmids and the presence of mutations were confirmed using DNA sequencing.

### 4.2. Protein Expression and Purification

To obtain the recombinant HA3-NTNHA (rHA3-NTNHA) complexes, *Escherichia coli* Rosetta2 (DE3) cells (Merck) transformed with the co-expression plasmids were grown in Terrific Broth (TB) media. Protein expression was induced using the Overnight Express^TM^ Autoinduction System 1 (Merck) at 18 °C for 48 h. The cells were then harvested and lysed in a lysis buffer (50 mM Tris-HCl, pH 7.4, and 300 mM NaCl) by sonication. The His-tagged proteins were bound to HisTrap HP (Cytiva, Marlborough, MA, USA) and eluted with His elution buffer (50 mM Tris-HCl pH 7.4, 300 mM NaCl, 300 mM imidazole). To purify the HA3-NTNHA complex, the eluates were loaded onto StrepTrap HP (Cytiva) equilibrated with PBS (pH 7.4), and the bound proteins were eluted with 3 mM D-desthiobiotin.

Recombinant FLAG-tagged HA1 serotypes B and A (rHA1/B-FLAG and rHA1/A-FLAG, respectively) and recombinant FLAG-tagged HA2 serotypes B and A (FLAG-rHA2/B and FLAG-rHA2/A, respectively) were prepared as previously described [13]. Native BoNT serotype B-Okra (nBoNT/B) was prepared as described previously [41].

All proteins were dialyzed against PBS (pH 7.4 or 6.0) and stored at −80 °C until needed. Protein concentrations of the samples were determined using a Pierce BCA assay (Thermo Fisher Scientific).

### 4.3. In Vitro Reconstitution and Purification

To reconstitute recombinant HA complexes (rHA: HA1 + HA2 + HA3), recombinant HA1, HA2, and HA3 proteins were mixed at a molar ratio of 4:4:1 in PBS (pH 7.4). To reconstitute recombinant NAP complexes (rNAP: NTNHA + HA), recombinant HA1, HA2, and HA3-NTNHA were mixed at a molar ratio of 12:12:1 in PBS (pH 7.4). The mixtures were then incubated at 37 °C for 3 h. The complexes were purified using StrepTrap HP and dialyzed against phosphate-buffered saline PBS (pH 6.0).

To reconstitute recombinant L-PTC (rL-PTC), nBoNT/B and rNAP proteins were mixed at a molar ratio of 3:1 in PBS (pH 6.0) and incubated at 37 °C for 3 h. The rL-PTCs were bound to a-Lactose gels (EY laboratories, San Mateo, CA, USA) equilibrated with PBS (pH 6.0) and then eluted with 0.2 M lactose. Purified proteins were dialyzed against PBS (pH 6.0). The protein concentrations of the samples were determined using a Pierce BCA assay.

### 4.4. Size Exclusion Chromatography

10 μg of each protein (rHA1/B, rHA/2, rHA3, rNTNHA/B, rNTNHA/B-rHA3/B, rHA/B, rNAP/B, native BoNT/B, rL-PTC/B-WT, and native L-PTC/B) was loaded onto a Superdex 200 Increase 10/300 GL column (Cytiva) equilibrated with PBS (pH 6.0) using ÄKTA pure (Cytiva). Elution was performed with the same buffer at a flow rate of 0.75 mL/min and monitored by the absorbance at 280 nm.

### 4.5. Western Blotting

Equimolar amounts of protein (native L-PTC/B, rL-PTC/B-WT, rL-PTC/B-KA, rL-PTC/BA-WT, and rL-PTC/BA-KA) were separated by 15% SDS-PAGE and transferred onto Immobilon-P^TM^ PVDF membranes (Merck). After blocking with 5% skim milk, the membranes were incubated with antibodies against BoNT/B (rabbit, anti-serum [15]), His tag (mouse monoclonal Ab, clone OGHis, MBL, Tokyo, Japan), Strep-tag II tag (mouse monoclonal Ab, Cat. 71590-3, Merck), and FLAG tag (mouse monoclonal Ab, clone M2, Merck), followed by appropriate HRP-conjugated secondary antibodies (Jackson ImmunoResearch, West Grove, PA, USA). Subsequently, the membranes were developed using ECL Select (Cytiva) and visualized with an ImageQuant^TM^ LAS 4000 mini (Cytiva).

### 4.6. Mucin ELISA

96-well ELISA plates (IWAKI, Tokyo, Japan) were coated with 100 ng/mL porcine gastric mucin (PGM; M1778, Sigma, Darmstad, Germany) and bovine submaxillary mucin (BSM; M3895, Sigma) at 37 °C for 1 hr. After blocking with 1% BSA/PBS-T (pH 6.0), the plates were incubated with 10 nM L-PTCs or HAs at 37 °C for 1 hr. After washing with PBS-T (pH 6.0), the plates were probed with antibodies against BoNT/B or FLAG tags, followed by incubation with the appropriate HRP-conjugated secondary antibodies. Subsequently, the plates were developed using ABTS (Merck), and the absorbance at 405 nm was measured.

### 4.7. Transepithelial Electrical Resistance (TER) Assay

TER was measured using a Millicell-ERS (Merck), as described previously [15]. Briefly, Caco-2 cell monolayers were established on Transwell^TM^ filters (Corning, Glendale, AZ, USA) and 10 nM L-PTCs or HAs were added to the basolateral chambers. TER was measured at different time points up to 48 h post-addition.

### 4.8. Mouse Bioassay

Female BALB/c mice aged 7–8 wk were purchased from Japan SLC, (Shizuoka, Japan). The mice were fasted for 4 h before the challenge. For intraperitoneal (i.p.) administration, the mice were injected i.p. with 100 pg of rL-PTCs in 300 μL of bioassay buffer (0.1% gelatin, 10 mM NaPi, pH 6.0). For intragastric (i.g.) administration, the mice were gavaged i.g., with 200 ng of rL-PTCs/B or 2 μg of rL-PTCs/BA in 300 μL of bioassay buffer. The mice were re-fed 1 h after the challenge, and the survival rate was assessed every 12 h over the subsequent five days. The toxins were administered in a single-blind manner, and intoxication was scored by two investigators.

### 4.9. Statistical Analysis

Statistical testing was performed using RStudio (R version 4.1.2) and Prism 10 (GraphPad Software, Boston, MA, USA). Statistical significance was evaluated using one- or two-way ANOVA with Tukey’s HSD post hoc test. Differences with *p* < 0.05 were considered statistically significant.

## Figures and Tables

**Figure 1 toxins-17-00443-f001:**
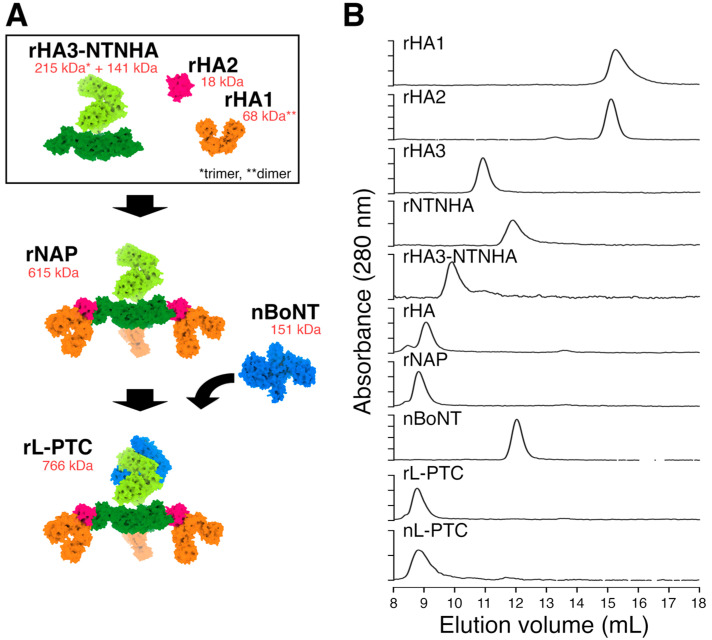
In vitro reconstitution of recombinant large progenitor toxin complex derived from *C. botulinum* serotype B-Okra (rL-PTC/B). (**A**) Schematic model of in vitro reconstitution of rL-PTC, a 766 kDa complex composed of nBoNT, rNTNHA, rHA3, rHA2, and rHA1 in a stoichiometry 1:1:3:3:6. (**B**) Size exclusion chromatography analysis. Recombinant proteins (rL-PTC/B-WT, rNAP/B, rNTNHA/B, and rHAs/B) and native proteins (nL-PTC/B and nBoNT) were subjected to size exclusion chromatography on a Superdex 200 Increase 10/300 GL column. Protein elution profiles were monitored at 280 nm. BoNT, botulinum neurotoxin; NTNHA, non-toxic non-hemagglutinin; HA, hemagglutinin; NAP, neurotoxin-associated protein; rL-PTC, recombinant L-PTC; nL-PTC, native L-PTC; nBoNT, native BoNT. Asterisks indicate the molecular weights of rHA3 as a trimer (*) and rHA1 as a dimer (**).

**Figure 2 toxins-17-00443-f002:**
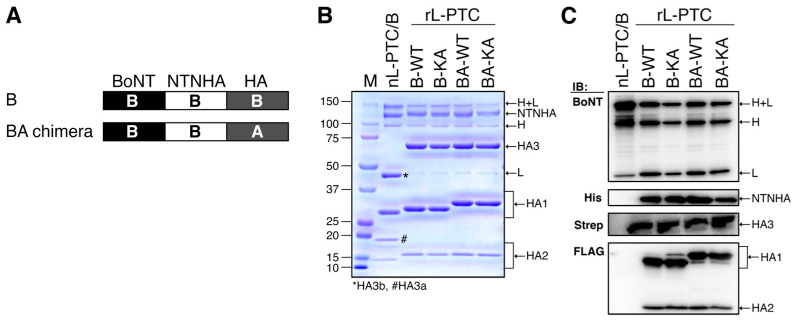
The molecular compositions of the purified rL-PTCs. (**A**) Schematic diagram of rL-PTC serotype B-Okra (rL-PTC/B-WT) and chimeric rL-PTC (rL-PTC/BA-WT) comprising BoNT/B, NTNHA/B, and HA derived from A-62A (HA/A). The purified nL-PTC/B, rL-PTCs/B, and rL-PTCs/BA were analyzed by SDS-PAGE with Coomassie brilliant blue staining (**B**) and Western blotting (**C**). WT: wild type, KA: HA3-K607A mutant. The membranes were probed with anti-BoNT/B antiserum, anti-His tag, anti-Strep tag II, and anti-FLAG tag antibodies, followed by the corresponding horseradish peroxidase-conjugated secondary antibodies. Asterisk (*) and sharp sign (#) indicate C-terminal (HA3b) and N-terminal (HA3a) domains of HA3, respectively. H + L, intact BoNT (heavy chain and light chain); H, heavy chain of BoNT; L, light chain of BoNT; M, marker.

**Figure 3 toxins-17-00443-f003:**
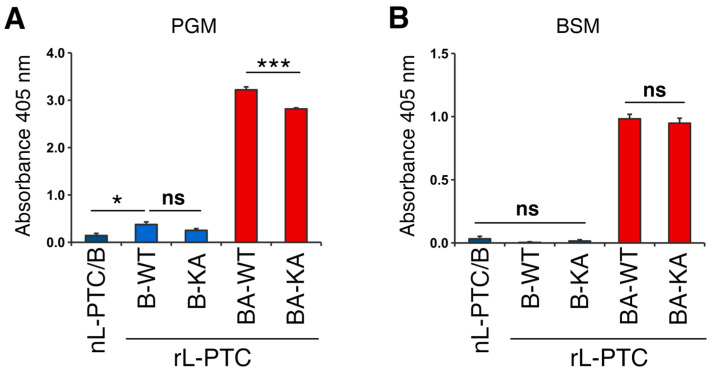
Carbohydrate-binding activities of rL-PTCs. HA-mediated binding of L-PTCs to porcine gastric mucin (PGM, **A**) and bovine submaxillary mucin (BSM, **B**). 10 nM L-PTCs were added to PGM- or BSM-coated ELISA plates, and then the plates were probed with anti-BoNT/B anti-serum. Data are representative of at least four independent experiments. Values are mean ± S.E. of triplicate wells. The data were analyzed by one-way ANOVA with Tukey’s HSD post hoc test (* *p* < 0.05, *** *p* < 0.001, ns: not significant).

**Figure 4 toxins-17-00443-f004:**
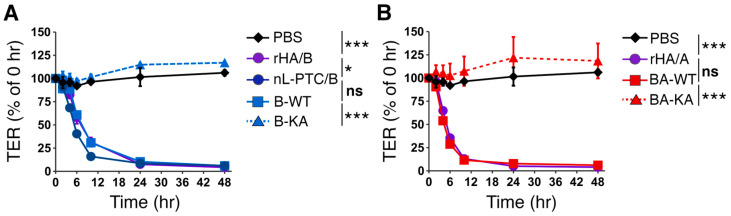
Barrier-disrupting activities of rL-PTCs. 10 nM L-PTCs (nL-PTC/B; rL-PTC/B-WT, B-WT; rL-PTC/B-KA, B-KA; rL-PTC/BA-WT, BA-WT; rL-PTC/BA-KA, BA-KA) and rHAs (rHA/B, rHA/A) ((**A**): serotype B-Okra, (**B**): serotype A-62A and rL-PTCs/BA) were added to the basolateral chamber of the Caco-2 cell monolayers. Transepithelial electrical resistance (TER) was measured at 0, 2, 4, 6, 10, 24, and 48 h after treatment. Data are representative of at least two independent experiments. Values are mean ± S.E. of duplicate (nL-PTC/B) or triplicate (the others) wells. The data were analyzed by two-way ANOVA with Tukey’s HSD post hoc test (* *p* < 0.05, *** *p* < 0.001, ns: not significant).

**Figure 5 toxins-17-00443-f005:**
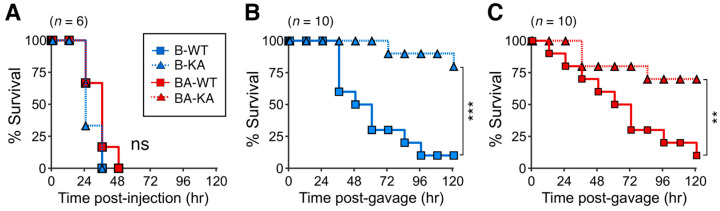
Intraperitoneal and oral toxicities of rL-PTCs in mice. (**A**) Female BALB/c mice were injected i.p. with 100 pg rL-PTCs (*n* = 6/group). (**B**,**C**) Female BALB/c mice were gavaged with 200 ng rL-PTCs/B (*n* = 10/group, (**B**)) and 2 μg r-L-PTCs/BA (*n* = 10/group, (**C**)). ns: not significant, ** *p* < 0.01, *** *p* < 0.001; by log-rank test.

## Data Availability

The original contributions presented in this study are included in the article/Appendix A. Further inquiries can be directed to the corresponding author(s).

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
