# Peer review of "Botulinum Toxin Complex Serotype B-Okra Exerts Systemic Toxicity via the Oral Route by Disrupting the Intestinal Epithelial Barrier"

_toxins, 2025, doi:10.3390/toxins17090443_

Round 1

Reviewer 1 Report

Comments and Suggestions for Authors

The authors describe their assessment to demonstrate that BoNT serotype B-Okra disrupts the epithelial barrier of enterocytes and exerts oral toxicity. I have a few comments and questions:

Introduction

Line 30, to form medium PTC (M-PTC) …

Line 32, M-PTC

Line 59, FITC shall be explained

Results

Line 91, caption Figure 2C, it should be briefly described how the blotted proteins were probed.

Figure 2C on page 3 and as separate Figure, the respective anti-BoNT and the anti-FLAG images seem to be identical, which does not seem to be the case for the respective anti-His and anti-Strep images (might be due to the fact that on page 3, for these two images, only a small section of the blot is depicted.

Line 101, Figure 3B and C and text in line 99 (“similar binding affinities”); how were the assays calibrated? The same concentration of the two mucins was used, but the absorbance values are different among the mucins; can it be concluded that the carbohydrate binding activity of BA-WT and BA-KA towards BSM is higher?

Line 131, at least 10-fold higher; how can this be deduced from Figure 5B/C? Which time point was taken to come to this conclusion? Why was the administered amount 10-fold higher for rL-PTCs/BA compared to rL-PTCs/BB?

Materials and Methods

Lines 208-209, the description is not fully clear, are these two approaches or only one? If two, which what were the recombinant HA complexes mixed?

Line 246, For intraperitoneal administration,

Line 254, ethics statement needs to be added

Reviewer 2 Report

Comments and Suggestions for Authors

The article, “Botulinum Toxin Complex Serotype B-Okra Exerts Systemic Toxicity Via the Oral Route by Disrupting the Intestinal Epithelial Barrier.”  This manuscript describes the mechanism by which BoNT/B Okra induces toxicity.  While this manuscript describes some very interesting results and has depicts the data within very clear images, excessive  use of acronyms/abbreviations and in some areas, poor sentence construction detract from the content.  As a result, full evaluation of the scientific merit of this manuscript was not possible.

1-The goal of a manuscript is to disseminate and share information. To achieve that goal, the manuscript must be READABLE.  Please minimize the use of acronyms—your manuscript might be slightly longer, but it will be far more readable.  Only use acronyms if a particular phrase is used >3x throughout the manuscript. Define the acronym the first time it is used.  Avoid use of non-standard acronyms—better to write out even if used >3x within manuscript.  You want your reader to read and comprehend the information provided; you do not want your reader to have to repeatedly figure out what each acronym means.

              -Write out NAPs throughout manuscript.

              -Write out NTNHA throughout manuscript.

              -Write out SEC.

              -Write out CBB.

              -Write out M-PTC and L-PTC.

2-Lines 7-10.  Take care with sentence construction. Also avoid use of uderscores in abbreviations/acronyms. Avoid excessive use of acronyms.  See below.

Rewrite:  “BoNTs associate with neurotoxin-associated proteins (NAPs) to form a medium progenitor toxin complex (PTC) (M-PTC) and large PTC (L-PTC). The L-PTC serotype A-62A (L-PTC/A_62A) specifically targets intestinal M cells for invasion, whereas L-PTC serotype B-Okra (L-PTC/B_Okra), which exhibits higher toxicity via the oral route, is mainly taken up by enterocytes.”  

Example:  “BoNTs associate with neurotoxin-associated proteins to form a medium and large progenitor toxin complex. The large progenitor toxin complex A-62A (L-PTC/A-62A) specifically targets intestinal M cells for invasion, whereas large progenitor toxin Serotype B-Okra (L-PTC/B-Okra), is mainly taken up by enterocytes and exhibits higher toxicity via the oral route.”

3-Line 15, “Our results demonstrate that the toxin of B-Okra...” Rewrite to: “Our results demonstrate that the B-Okra toxin...”

4-Line 23, “Botulinum toxin (BoNT) is the most potent toxin that causes botulism....” BoNT is the only toxin that causes botulism so rewrite to: “Botulinum toxin is the most potent naturally occurring toxin and is responsible for causing botulism...”

5-Line 28, “NAPs comprise a non-toxic...”  Rewrite to:  “Neurotoxin associated proteins are comprised of a non-toxic....”

6-Line 30, no definition for NHNHA.  Better to write out—this is not a standard abbreviation and due to it’s length it is very difficult to remember what it stands for.

7-Line 35, “....barrier-disrupting...” Rewrite to:  “...barrier-disruption...”

8-Line 59, “increased” should be “increases”.

9-Line 56, begin a new paragraph with, “A single...” and end this new paragraph with the information included within the paragraph that begins on Line 62.

10-Lines 70, 71 and 74, choose one set of abbreviations—either use rL-PTC/B, rL-PTC/B and HA3-K607A or rL-PTC/BB-WT, rL-PTC/BA-WT,  rL-PTC/BB-KA and rL-PTC/BA-KA.  Don’t include both sets of labels for each entity (preferably use the shorter version of each as that is how these entities are referred to within the abstract and images).

11-Figure #2, within legend, asterisk and sharp sign are referred to.  Include in parentheses the symbols for these, i.e. “Asterisk (*) and sharp (#) sign...” Also include “respectively” at end of sentence. “Asterisk (*) and sharp (#) sign indicate C-terminal (HA3b) and N-terminal (HA3a) domains of HA3, respectively.”

12-Figure legends, some of the abbreviations used within the images are defined and some are not; indicate what each abbreviation within the images represent.

13-Figure 4, what is “TER” on y-axes?

14-The authors present several very nice images showing their data.  The Results section should describe what the results are—simply referring to the images is not adequate.

15-While the Discussion can summarize your findings it should really relate your findings to those of others.  The second paragraph of the Discussion appears to refer to work this group previously conducted.  How do the findings described in the current manuscript relate to the findings of other groups?  Do not refer to figures.  Final paragraph of Discussion is good.

16-Methods should provide adequate detail that someone else could replicate your studies.  4-4-What proteins were subjected to size exclusion chromatography?  How was size exclusion chromatography conducted? What proteins were subjected to SDS-PAGE and Western blotting?

17-Statement regarding biosafety/regulatory protocols for work (particularly recombinant work) with Clostridium botulinum and its nucleic acids?

18-Authors should have a scientist outside the area of study carefully read the manuscript for clarity and consistency.

Reviewer 3 Report

Comments and Suggestions for Authors

This manuscript presents a well-structured and methodologically rigorous study addressing the role of epithelial barrier-disrupting activity in the high oral toxicity of the hyper-oral-toxic (HOT) type L-PTC/B-Okra. By generating recombinant toxin complexes and site-directed mutants, the authors provide compelling evidence that disruption of the intestinal epithelial barrier, mediated by HA binding to E-cadherin, is critical for oral toxicity. The experimental approach is logically designed, and the data are presented clearly.

  1. The manuscript contains numerous recombinant constructs and variants (e.g., rL-PTC/BB-WT, rL-PTC/BA-WT, rL-PTC/BB-KA, rL-PTC/BA-KA, native complexes, and various HA proteins), which may be difficult for readers to follow when reading through the Results and Methods. It is strongly recommended to include a schematic figure summarizing all constructs used in the study, indicating their component proteins, source serotypes, and key mutations. This would help readers quickly understand the experimental design.

  1. The current study evaluates oral toxicity using a single intragastric dose for each toxin variant. While these data demonstrate qualitative differences in lethality between WT and KA mutants, they do not provide a quantitative measure of the magnitude of toxicity reduction. Without dose–response information, it is difficult to compare the relative potency of the variants or to determine whether the observed differences are proportional or threshold effects. It should be better to perform oral toxicity assays at multiple doses for each variant and determine the median lethal dose (LDâ‚…â‚€) values for both WT and KA mutants.

  1. In Figures 3 and 4, the authors present mean/S.E. values from multiple independent experiments; however, statistical analysis should be performed to determine whether the observed differences (or similarities) between groups are statistically significant.

  1. Ensure consistent nomenclature (e.g., “rL-PTC/BB-KA” vs. “rL-PTCs/BB-KA”, and Line 85 “r-LPTCs”)

Round 2

Reviewer 2 Report

Comments and Suggestions for Authors

The revised version of this manuscript reflects great improvement relative to the initial version.

Section 2.4:

Lines 179 & 180-"We assessed the BoNT toxicity of rL-PTCs by intraperitoneal (i.p.) administering 200 179 pg of rL-PTCs to mice." Revise to, "The intraperitoneal toxicity of rL-PTCs was assessed by intraperitoneal (i.p.) administration of 200 pg rL-PTCs to mice."

Line 186-"The oral toxicity of rL-PTCs...." Begin new paragraph with this sentence.

Figure 5 legend-"Mouse intraperitoneal and oral toxicities of rL-PTCs." Revise to, "Intraperitoneal and oral toxicity of rL-PTCs in mice."

Author Response

The revised version of this manuscript reflects great improvement relative to the initial version.

Section 2.4:

Lines 179 & 180-"We assessed the BoNT toxicity of rL-PTCs by intraperitoneal (i.p.) administering 200 179 pg of rL-PTCs to mice." Revise to, "The intraperitoneal toxicity of rL-PTCs was assessed by intraperitoneal (i.p.) administration of 200 pg rL-PTCs to mice."

We have revised it. (L146-147)

Line 186-"The oral toxicity of rL-PTCs...." Begin new paragraph with this sentence.

We have revised it. (L154)

Figure 5 legend-"Mouse intraperitoneal and oral toxicities of rL-PTCs." Revise to, "Intraperitoneal and oral toxicity of rL-PTCs in mice."

We have revised it. (L162)

Reviewer 3 Report

Comments and Suggestions for Authors

The revised manuscript has significantly improved and now clearly resolves the concerns raised.

Author Response

The revised manuscript has significantly improved and now clearly resolves the concerns raised.
We thank the reviewer for their positive evaluation and are pleased that the revised manuscript has addressed the concerns.